# *Bacillus anthracis’* PA_63_ Delivers the Tumor Metastasis Suppressor Protein NDPK-A/NME1 into Breast Cancer Cells

**DOI:** 10.3390/ijms21093295

**Published:** 2020-05-06

**Authors:** Ina Felix, Santosh K. Lomada, Holger Barth, Thomas Wieland

**Affiliations:** 1Institute of Pharmacology and Toxicology, University Ulm Medical Center, 89081 Ulm, Germany; ina.felix@uni-ulm.de; 2Experimental Pharmacology, European Center for Angioscience, Mannheim Medical Faculty, Heidelberg University, 68167 Mannheim, Germany; Santosh.Lomada@medma.uni-heidelberg.de; 3DZHK (German Center for Cardiovascular Research), Partner Site, Heidelberg/Mannheim, Germany

**Keywords:** tumor metastasis suppressor protein, NDPK-A, NME1, Nm23-H1, anthrax toxin, protein transport, targeted drug delivery, protective antigen PA_63_

## Abstract

Some highly metastatic types of breast cancer show decreased intracellular levels of the tumor suppressor protein NME1, also known as nm23-H1 or nucleoside diphosphate kinase A (NDPK-A), which decreases cancer cell motility and metastasis. Since its activity is directly correlated with the overall outcome in patients, increasing the cytosolic levels of NDPK-A/NME1 in such cancer cells should represent an attractive starting point for novel therapeutic approaches to reduce tumor cell motility and decrease metastasis. Here, we established the *Bacillus anthracis* protein toxins’ transport component PA_63_ as transporter for the delivery of His-tagged human NDPK-A into the cytosol of cultured cells including human MDA-MB-231 breast cancer cells. The specifically delivered His_6_-tagged NDPK-A was detected in MDA-MB-231 cells via Western blotting and immunofluorescence microscopy. The PA_63_-mediated delivery of His_6_-NDPK-A resulted in reduced migration of MDA-MB-231 cells, as determined by a wound-healing assay. In conclusion, PA_63_ serves for the transport of the tumor metastasis suppressor NDPK-A/NME1 into the cytosol of human breast cancer cells In Vitro, which reduced the migratory activity of these cells. This approach might lead to development of novel therapeutic options.

## 1. Introduction

Metastasis is the leading cause of mortality in cancer patients, even if the primary tumor has been removed. Due to the complexity of the processes in metastasis major breakthroughs in inhibiting this often deadly spreading of cancer cells have not been achieved so far [1,2]. Metastasis suppressors are proteins which exert no inhibitory effect on primary tumor growth, but significantly reduce metastasis. Thus, specifically increasing the amount of such a protein present in highly metastatic tumor cells might offer an opportunity to interfere with these processes [3,4].

The human *nm23-H1* gene, now more generally named *nme1*, was the first metastasis suppressor identified [5]. By comparison with its *D. melanogaster* orthologue, *awd*, it became evident that it encoded the A isoform of the enzyme nucleoside diphosphate kinase (NDPK-A, also named NME1) [6]. In mammals, the *nme* family consists of 10 genes, although the gene products NME1 and NME2 members have been studied with regard to metastasis in more detail. Overexpression of, for example, NDPK-A/NME1 in metastatic tumor cell lines significantly reduced In Vivo metastasis with no effect on primary tumor size [7]. In In Vitro experiments performed in a variety of tumor cells, it was shown that NDPK-A/NME1 re-expression reduced the migration in Boyden chamber as well as wound healing assays stimulated with multiple attractants, which suggest a central role in the regulation of tumor cell motility [8,9,10]. Recent data showed that dynamin 2 oligomerization is promoted by NDPK-A/NME1 in breast cancer cells. As dynamin oligomerization is required for endocytosis of, e.g., chemotactic EGF receptors and others, the enhancement of the internalization of such receptors by NDPK-A/NME1 might be part of the underlying mechanism [11].

Thus, its metastasis-suppressing function in certain breast cancer types, especially those with a decreased level in NDPK-A/NME1 expression, is well established and at least partially understood. Therefore, in such cancers the restauration of NDPK-A levels in the cells should be beneficial and the targeted delivery of enzymatically active human NDPK-A into these cells an attractive starting point for the development of novel therapeutic options.

However, the delivery of therapeutic proteins or peptides into the cytosol of mammalian cells is a major challenge in pharmacology because transport across cell membranes is required. In recent years, non-toxic mutants or portions of bacterial protein toxins, which are nature’s best transporter molecules, were exploited by various groups including our own for this purpose [12,13,14,15,16]. These toxins enter mammalian cells by receptor-mediated endocytosis and deliver an enzymatically active subunit from acidic endosomal vesicles into their cytosol [14]. There, this enzyme modifies its specific cellular substrate molecule which interferes with the structure and/or function of the cell, thereby causing severe diseases such as botulism, tetanus, diphtheria or anthrax. For this unique mode of action, these toxins have a particular structure: they consist of three functionally different subunits, which enable firstly receptor-binding on the cell surface (B-subunit), then the transport of the catalytic subunit across endosomal membranes (T-subunit) and finally, the enzyme modification by the enzymatically active A-subunit. For some of these ABT-toxins it was demonstrated by us and others that their B/T-subunits can deliver “foreign” proteins in to the cytosol instead of their natural A-subunit [12,14,16,17].

A well-established, toxin-based transporter is the B/T-subunit of the anthrax toxins from *Bacillus anthracis*, namely protective antigen PA_63_. Anthrax toxin, which is the causative agent of the disease anthrax, is composed of the central B7T-subunit PA_63_ (63 kDa), which delivers two different and separate enzyme subunits into the cytosol of human cells, the lethal factor LF, an adenylyl cyclase, and the edema factor EF, a protease [18]. For intoxication of cells, PA_63_ binds to specific receptor molecules on target cells and forms oligomers (heptamers or octamers) at the cell surface [18]. Then LF or EF binds to the cell-bound PA_63_ oligomers and the toxin complexes are internalized into endosomes by receptor-mediated endocytosis. Following acidification of the endosomes by a vesicular v-ATPase, the PA_63_ oligomers change their conformation and insert as trans-membrane pores into the endosomal membrane [19,20,21]. Finally, LF or EF translocate through these pores from the endosomal lumen into the host cells cytosol where they exhibit their cytotoxic enzyme activities [18,22]. This sophisticated mechanism can be exploited for delivery of “foreign” proteins including enzymes into the cytosol of mammalian cells. It was demonstrated that PA_63_ delivers proteins fused either to the N-terminal domain of LF [23,24,25,26] or to a short sequence of positively charged amino acids, such as histidine residues [27,28,29]. This allows for the PA_63_-mediated transport of proteins that contain a conventional N-terminal His_6_-tag, which usually serves for affinity purification of recombinant proteins [30,31]. By using this approach, more than 30 proteins and peptides were successfully delivered into the cytosol of cells via PA_63_ [29], including the enzyme domain DTA of diphtheria toxin. His_6_-DTA as a model cargo [27] is also of particular interest for this present study because it enables monitoring the transport of a His-tagged protein into the cytosol: DTA mono-ADP-ribosylates the elongation factor EF-2 in the cytosol of target cells, which inhibits protein synthesis in these cells and results in cell-rounding and cell death, specific endpoints that can be easily detected [32].

Here, we aimed to develop the PA_63_ transporter for delivery of recombinant human His_6_-NDPK-A into the cytosol of human breast cancer cells in order to increase the cytosolic levels of NDPK-A in such cancer cells and reduce their motility. The human breast carcinoma cell line MDA-MB-231 was used as model in this study because these cells are highly metastatic and show relatively low basal expression levels of endogenous NDPK-A/NME1. In conclusion, the results demonstrate that His_6_-NDPK-A is delivered into MDA-MB-231 cells by PA_63_-mediated transport, which reduced migration of these breast cancer cells.

## 2. Results

### 2.1. Purification and Characterization of Recombinantly Expressed His_6_-NDPK-A

His_6_-NDPK-A was recombinantly expressed in *E. coli* BL21 and purified via affinity chromatography. The identity of the purified His_6_-NDPK-A protein was confirmed by Western blotting with a specific antibody directed against the amino acid residues 134-152 of human NDPK-A (Figure 1B). To verify the enzyme activity of the purified His_6_-NDPK-A, its intrinsic nucleoside diphosphate kinase activity was analyzed by Western blotting. After performing an autophosphorylation assay, the 1-phosphohistidine-specific antibody confirmed the presence of the enzyme intermediate (Figure 1C). Furthermore, the His_6_-NDPK-A-catalyzed conversion of ADP to ATP was quantified by In Vitro kinase assay. The result shows a concentration-dependent increase of the luminescence signal and the control reactions revealed the substrate dependency (Figure 1D). Taken together, the recombinant His_6_-NDPK-A was enzymatically active and could be used in further experiments to analyze its transport into cells via the PA_63_ transporter.

### 2.2. MDA-MB-231 Cells are Susceptible to PA_63_-Mediated Delivery of His-Tagged Protein Cargo

To investigate whether His_6_-NDPK-A can be delivered into the cytosol of MDA-MB-231 breast cancer cells via the PA_63_ transporter, we first confirmed that this cell line is susceptible for the PA_63_-mediated protein transport using a well-established His_6_-tagged cargo protein, His_6_-DTA, the recombinant enzyme domain of diphtheria toxin (DT). When His_6_-DTA reaches the cytosol, it ADP-ribosylates elongation factor EF-2 and thereby inhibits protein synthesis. This results in cell rounding, which can be easily monitored by microscopic methods and represents a sensitive and specific endpoint to monitor the transport of enzymatically active His_6_-DTA into the cytosol. His_6_-DTA was applied to the cells in presence or absence of PA_63_ and the DTA-induced cell rounding analyzed (Figure 2A). The morphological changes were obvious after 4 h and even more evident after 24 h. The changes in cell morphology were only observed after application of the combination of His_6_-DTA plus PA_63_ but not after treatment with His_6_-DTA alone, clearly indicating the specific PA_63_-mediated delivery of His_6_-DTA into the cytosol of MDA-MB-231 cells. Moreover, the treatment of the cells with bafilomycin A1 (BafA1), which inhibits endosomal acidification, prevented the changes of cell morphology after treatment with His_6_-DTA plus PA_63_, confirming the specific PA_63_-mediated transport. However, MDA-MB-231 cells responded slower to PA_63_/His_6_-NDPK-A than HeLa cells (Figure 2B), where rounding was complete after 4 h. Therefore, the incubation times for the next experiments where adjusted accordingly and HeLa cells were tested in parallel to the MDA-MB-231 cells. For both cell lines, these results were confirmed by quantitative analysis of rounded cells. In conclusion, the results clearly showed that this breast cancer cell line can be used to investigate the transport of His-NDPK-A via PA_63_.

### 2.3. PA_63_ Mediates Delivery of His_6_-NDPK-A into Cells

Prompted by the finding that PA_63_ serves for delivery of His_6_-tagged DTA into the cytosol of MDA-MB-231 cells, we next tested whether His_6_-NDPK-A can be introduced into these cells by this approach. To this end, we first analyzed the binding of His_6_-NDPK-A in the absence and presence of PA_63_ to MDA-MB-231 (Figure 3A) and HeLa cells (Figure 3B). After incubation of the cells at 4 °C, where binding but not uptake occurs, with His_6_-NDPK-A in presence or absence of PA_63_, the amounts of cell-bound His_6_-NDPK-A were analyzed by Western blotting. For both cell lines, there was some unspecific binding of His_6_-NDPK-A in the absence of the transporter. However, for both cell lines, the amount of cell-bound His-NDPK-A clearly increased in the presence of PA_63_ (Figure 3), indicating a specific binding of His-NDPK-A to the cells via PA_63_. HeLa cells bound more His-NDPK-A via PA_63_, which might be due to higher expression of anthrax toxin receptor in this cell line.

Having confirmed the PA_63_-mediated binding of His_6_-NDPK-A to MDA-MB-231 cells, the delivery of His_6_-NDPK-A via PA_63_ into these cells was analyzed. Here, His_6_-DTA was again included as a control because the PA-mediated transport into the host cell cytosol can be easily monitored in terms of cell rounding. For the analysis via immunofluorescence microscopy, MDA-MB-231 cells were incubated for 48 h at 37 °C with His_6_-NDPK-A in the absence or presence of PA_63_, and His_6_-NDPK-A was detected with a specific antibody against the His-tag and Alexa488-conjugated secondary antibody (Figure 4A). In parallel, the same approach was performed with His_6_-DTA instead of His_6_-NDPK-A. The results show an increased signal for both His_6_-tagged cargo proteins (green) in the presence of PA_63_, clearly indicating the specific PA_63_-dependent transport into the cells. Moreover, cells treated with His_6_-DTA plus PA_63_ rounded up, indicating that His_6_-DTA reached the cytosol. Here, the signal was stronger compared to the cells treated with His_6_-NDPK-A plus PA_63_. The same set of experiments were performed with HeLa cells to exclude a cell-line specific effect. HeLa cells also showed the PA_63_-dependent transport of His_6_-NDPK-A and His_6_-DTA or (Figure 4B). Already after 3 h of incubation, the signal was obviously increased for His_6_-NDPK-a as well as His_6_-DTA when the transporter was present. Furthermore, co-staining of filamentous actin (red, bottom row) visualized the His_6_-DTA-induced cell-rounding.

To analyze the PA_63_-mediated transport of His_6_-NDPK-A in an alternative approach, cells were analyzed by Western blotting. As shown in Figure 5A, the difference in the molecular mass of the endogenous NDPK-A (17 kDa) and the recombinant His_6_-NDPK-A (18 kDa) taken up was clearly visible. There was no signal for His_6_-NDPK-A in untreated cells further confirming the identity of the introduced His_6_-NDPK-A protein. After incubation for 48 h with His_6_-NDPK-A in the presence of PA_63_, MDA-MB-231 cells showed a concentration-dependent increase of the amount of recombinant protein present. Interestingly, lower amounts of His_6_-NDPK-A were apparently taken up into the cells in the absence of PA_63_. Noteworthy, the introduction of His_6_-NDPK-A by PA_63_ into MDA-MB-231 cells did not impair cell viability at least within 48 h, as confirmed by the MTS test, monitoring mitochondrial activity (Figure 5B). As expected, the positive control DMSO (20% by vol.) reduced cell viability by about 50%.

Again, His_6_-DTA was used as a control cargo to monitor the cytosolic localization of His_6_-tagged proteins after delivery into MDA-MB-231 via PA_63_. Western blot analysis revealed His_6_-DTA uptake for cells treated with His_6_-DTA plus PA_63_, but not in cells treated with His_6_-DTA alone (Figure 5C). Accordingly, treatment of cells with BafA1 prevented the PA_63_-mediated transport of His_6_-DTA into the cells (Figure 5C). The presence of enzymatically active His_6_-DTA in the host cells cytosol was confirmed by sequential-ADP-ribosylation for the ADP-ribosylation status of the elongation factor EF-2 (Figure 5D). Lysates were incubated with fresh DTA and biotin-NAD and the amount of biotinylated, i.e., ADP-ribosylated EF-2, was determined by Western blotting. In contrast to all other conditions investigated, biotinylated EF-2 was detected in lysates from cells which were treated before with His_6_-DTA plus PA_63_, indicating that the EF-2 of these cells was already ADP-ribosylated by His_6_-DTA in the cytosol during the incubation of the still living cells. The presence of biotinylated EF-2 in all other samples indicates the absence of His_6_-DTA in the cytosol of the living cells prior to their lysis and the In Vitro assay (Figure 5D). Of note, also treatment of living cells with His_6_-DTA alone did not show any reduction, indicating that His_6_-DTA was not taken up into the cytosol of cells without PA_63_. Accordingly, treatment with BafA1 abolished the effect of His_6_-DTA plus PA_63_, indicating the specific uptake of His_6_-DTA via the PA_63_-dependent transport mechanism. In line with the EF-2 ADP-ribosylation data, the cell viability was significantly reduced after the 24-h incubation of cells with His_6_-DTA plus PA_63_ (Figure 5E). Again, this effect was prevented by BafA1. Treatment with staurosporine, a known inducer of apoptosis, reduced cell viability by about 50%.

Comparable results were obtained for HeLa cells (Figure 6), indicating that the findings are not restricted to a single cell line but have a broader implication. Again, the uptake of His_6_-NDPK-A into cells was clearly increased in the presence of PA_63_, as analyzed by Western blotting (Figure 6A), but did not exhibit adverse effects on the cell viability within at least 48 h of incubation (Figure 6B). The PA_63_ dependency was also observed for the uptake of His_6_-DTA and pre-incubation of cells with BafA1 blocked this uptake as analyzed in terms of His_6_-DTA-induced cell-rounding and Western blot detection of the internalized His_6_-DTA (Figure 6C). Moreover, only in the presence of PA_63_, His_6_-DTA reached the cytosol of cells as analyzed by sequential ADP-ribosylation of EF-2 (Figure 6D). In line with these results, cell viability was decreased when cells were treated with His_6_-DTA plus PA_63_ due to the His_6_-DTA activity in the cytosol (Figure 6E). However, even after 24 h, BafA1 prevented this effect, indicating the specificity of the His_6_-DTA transport into cells.

In conclusion, these results confirm a PA_63_-dependency for the delivery of His_6_-NDPK-A and His_6_-DTA into MDA-MB-231 and HeLa cells. For His_6_-DTA, the cytosolic localization as an active enzyme could be demonstrated due to its specific modification of the cytosolic substrate EF-2. Moreover, the PA_63_-mediated delivery of His_6_-NDPK-A is not cell-line specific.

### 2.4. Treatment with His_6_-NDPK-A Plus PA_63_ Decreased Migration of MDA-MB-231 Cells

Finally, it was investigated whether the PA_63_-mediated transport of the tumor metastasis suppressor protein His_6_-NDPK-A into the cytosol of MDA-MB-231 cells has an effect on the migration of these breast cancer cells. To this end, an In Vitro wound-healing assay was performed, where the cells were seeded into a dual chamber insert which was placed into a 24-well plate. When the cells were attached, the insert was removed thereby introducing the scratch. After removal of the detached cells by washing, His_6_-NDPK-A was applied in the presence or absence of PA_63_. In parallel, the same experiment was performed with His_6_-DTA instead of His_6_-NDPK-A. The migration of the MDA-MB-231-cells was monitored by microscopic analysis and photo-documentation. As shown in Figure 7A, there were less cells in the wound (scratch) area after 12 and 24 h when these cells were incubated with His_6_-NDPK-A or His_6_-DTA in the presence of PA_63_, suggesting a reduced migration of these cells. The representative pictures show a reduced closure of the scratch as soon as PA_63_ was present in addition to the His-tagged proteins. After 12 h, the scratch remained more open when cells were treated with either His_6_-NDPK-A or His_6_-DTA. The automated determination of the scratch area after 24 h in correlation to the scratch area at the start of the experiment confirmed this result (Figure 7B). There was a sustained significant reduction in migration of MDA-MB-231 cells after treatment with His_6_-NDPK-A plus PA_63_ after 12 h for up to 24 h. For this particular experiment, treatment with His_6_-DTA plus PA_63_ served as positive control. As expected, the migration of the treated cells was nearly abolished after 8 h and after 24 h cell-rounding was obvious. In line, a reduced cell viability occurred only in cells treated His_6_-DTA and PA_63_. In Figure 7D, scatter plots of three independent experiments are shown. The results corroborate the single experiment. The treatment of cells with His_6_-NDPK-A plus PA_63_ resulted in larger remaining scratch areas after 12 and 24 h compared to untreated cells. Therefore, the results implicate a PA_63_-mediated cytosolic delivery of His_6_-NDPK-A, which decreases the migration of MDA-MB-231 breast cancer cells In Vitro.

## 3. Discussion

With the results presented here, we provided proof-of-concept that the transport component of anthrax toxin PA_63_ serves for delivery of the tumor metastasis suppressor protein NDPK-A into human cells including breast cancer cells In Vitro. We used the established approach to fuse a common His_6_-tag to NDPK-A and purified and characterized this recombinant protein, as depicted in Figure 8.

Delivery of His_6_-NDPK-A by PA_63_ was confirmed by a set of experiments using different approaches. However, there was also some non-specific binding of His_6_-NDPK-A to the cells in the absence of PA_63_. In the presence of the transporter, this binding was obviously increased. Moreover, it could be confirmed that the transport of His_6_-NDPK-A occurred specifically via the PA_63_ uptake mechanism as it was prevented when cells were pretreated with BafA1 to inhibit endosomal acidification.

One main advantage of this approach is the intrinsic T-domain of PA_63_ for cytosolic delivery of the cargo proteins, which is a bottle neck of most available cellular transport systems including cell-penetrating peptides. Such peptides and their fused cargo proteins are efficiently internalized into endosomal vesicles but have limited capacity to translocate their cargo into the cytosol where it finds its cellular target substrate, i.e., drug target, for modification.

However, it is not possible to monitor the delivery of His_6_-NDPK-A into the cytosol by a direct approach. Therefore, we used the established His_6_-DTA as a model cargo for PA_63_-mediated transport into the cytosol in the same experiments in parallel to His_6_-NDPK-A. As the substrate for DTA, EF-2 is exclusively localized in the cytosol of cells and the DTA-catalyzed inactivation of EF-2 results in cell rounding and the simple microscopic analysis allows direct correlation of the cell rounding with the presence of enzymatically active His_6_-DTA in the cytosol. We took advantage of this approach to confirm that the breast cancer cell line MDA-MB-231 is sensitive for PA_63_-mediated protein delivery. Although these cells bound and internalized PA_63_-associated cargo proteins, they were less sensitive towards PA_63_ then HeLa cells. One reason might be that HeLa cells express more of the anthrax toxin receptors (CMG2/TEM8) on their surface. The uptake of His_6_-DTA into the cytosol was clearly confirmed. However, although His_6_-NDPK-A is a different protein with a different structure and unfolding/refolding abilities, its transport as His-tagged protein via the same PA_63_-dependent mechanism from endosomes into the cytosol might be an overall comparable process as observed for His_6_-DTA.

To prove whether NDPK-A reached the cytosol of MDA-MB-231 cells as an active enzyme, similar functional analysis was performed as after transfection-based overexpression of NME1/NDPK-A described before [11].

To this end, the migration of these cells was monitored over time in a wound healing/scratch assay. When the amounts of cells in the scratch area were monitored at different time points after generating the scratch, there were less cells present after treatment with PA_63_ plus His_6_-NDPK-A compared to cells treated with His_6_-NDPK-A alone or untreated cells. In contrast to His_6_-DTA uptake, increased delivery of His_6_-NDPK-A did not diminish the number of living cells, a cytotoxic effect of PA_63_ plus His_6_-NDPKA in terms of cell viability or division was excluded. Therefore, the reduced number of cells in the scratch likely results from the well described and mechanistically at least partially understood reduction in motility of MDA-MB-231 in which NME1/NDPK-A expression has been restored [8,9,10,11].

This suggests that indeed functional active NDPK-A reaches the cytosol of those cells and mediated this effect. In conclusion, as the anti-migratory effect of PA_63_ plus His_6_-NDPKA on MDA-MB-231 cells was apparently as effective as NME1/NDPK-A overexpression by conventional methods, our data provide proof-of-feasibility that the specific PA_63_-mediated introduction of a His_6_-tagged version of the tumor metastasis suppressor protein NME1/NDPK-A into the cytosol of cancer cells can be an attractive novel approach to reduce the migration and metastasis, e.g., of certain breast cancer types.

Future work must focus on the improvement of this system regarding efficiency and cell type-selectivity. Efficiency might be improved by fusing the N-terminal domain of LF (LF_N_) [23,24,25,33] to NDPK-A instead of the His_6_-tag or to use alternative amino acid tags instead of histidine residues [27,28,29]. It will also be important to treat other breast cancer cell lines as well as primary human breast cancer cells, which might express larger amounts of anthrax toxin receptors than the MDA-MB-231 cells, with PA_63_ plus His_6_-NDPKA or LF_N_-NDPK-A to investigate whether these cells show enhanced sensitivity towards this system.

Cell-type selectivity can be achieved by deleting the natural receptor binding capacity of PA_63_ by single amino acid exchange in its B-domain (mPA_63_) and fusing peptide ligands for receptors on the surface of certain cancer cells such as EGF or ZHER2, the ligand for HER2 receptor. This elegant approach was developed by the Collier group earlier and successfully applied in various studies In Vitro, ex vivo and In Vivo, without obvious immunological reactions caused by the PA-based delivery system [34,35,36,37,38]. Finally, the feasibility and efficacy of such approaches have to be validated in animal models for metastasis.

## 4. Material and Methods

### 4.1. Materials

Materials for cell culture were obtained from TPP (Trasadingen, Switzerland) or Sarstedt (Nümbrecht, Germany). Minimum Essential Medium (MEM), Dulbecco’s Modified Eagle Medium (DMEM), fetal calf serum (FCS), non-essential amino acids (NEAA, #11140-035) and sodium-pyruvate (Na-Pyr, #111360-070) as well as Penicillin/Streptomycin (#10378016) and Fungizone^®^ antimycotic (#15240062) were purchased from Life Technologies (Carlsbad, CA, USA).

### 4.2. Methods

#### 4.2.1. Cell Culture

Cells were cultured in medium containing 10% FCS, 1.5 g/L sodium bicarbonate, 2 mM L-glutamine, 1 mM sodium-pyruvate, 0.1 mM non-essential amino acids and 1% penicillin-streptomycin at 37 °C and 5% CO_2_. MEM was used for HeLa cells from DSMZ (CCL-2, Braunschweig, Germany) and DMEM for MDA-MB-231 cells from Cell Biolabs (HTB-26, San Diego, CA, USA). MDA-MB-231 cells were additionally supplemented with 0.25 µg/mL Fungizone^®^ antimycotic. If not mentioned otherwise, all experiments were performed in fully supplemented medium and cells were seeded one day prior to the experiment.

#### 4.2.2. Expression and Purification of Recombinant Proteins

The expression vector pD444-NH encoding for human *nm23-H1* including an N-terminal His_6_-tag (pD444-NH_His_6_-nm23-H1) was kindly provided by P. Attwood (Perth, Australia) to T. Wieland. After transformation into *Escherichia* (*E.*) *coli* BL21 DE3 and induction with 0.5-mM IPTG (#A1008, AppliChem, Darmstadt, Germany) at an optical density of 0.6–0.8 expression was performed for 3 h at 37 °C. Cells were harvested at 6.000 rcf for 15 min at room temperature. After a freeze–thaw cycle, the pellet was resuspended in a 10-mL lysis buffer (NaH_2_PO_4_ (50 mM), NaCl (300 mM), imidazole (10 mM), EDTA (1 mM), glycerol (5%), pH 8) per liter of the main culture. Lysis was performed for 30 min on ice with 1 mg/mL of lysozyme (#89833, Thermo Scientific, Waltham, MA, USA) followed by 10 cycles of sonication (40% intensity, 30 s each burst, halted for 30 s on ice in between). Lysate was cleared via centrifugation at 10,000 rcf for 30 min at 4 °C and subsequently filtered (Filtropur S0.45 and S0.2, Sarstedt, Nümbrecht, Germany). With the ÄKTA™ system (GE Healthcare Life Sciences, Uppsala, Sweden), His_6_-NDPK-A (expression product of *Nm23-H1*) was purified via Ni^2+^ (#745410, Macherey-Nagel, Düren, Germany) affinity chromatography. After binding, the column was washed with washing buffer (NaH_2_PO_4_ (50 mM), NaCl (300 mM), imidazole (20 mM), EDTA (1 mM), glycerol (5%), pH 8) and the protein eluted with an increasing gradient of elution buffer (NaH_2_PO_4_ (50 mM), NaCl (300 mM), imidazole (250 mM), EDTA (1 mM), glycerol (10%), pH 8). Buffer was exchanged to storage buffer (Tris-HCl (20 mM), DTT (1 mM), glycerol (10%), pH 7.5) with the Vivaspin^®^ 20 centrifugal concentrator, molecular weight cut-off (MWCO) of 5 kDa (#VS2011, Sartorius, Goettingen, Germany). Sodium dodecyl sulfate–polyacrylamide gel electrophoresis (SDS-PAGE) and protein determination by Bradford assay were used for quantification and protein identity was confirmed via immunoblotting and the use of the specific anti-Nm23-H1 antibody. The final concentration was determined at 1.018 mg/mL, total yield of ~1.7 mg His_6_-NDPK-A per liter main culture, protein was stored at −80 °C.

The expression vector encoding for His_6_-DTA (pET-15b_His_6_-DTA) was kindly provided by R.J. Collier (Boston, MA, USA) and transformed into *E.coli* BL21. Protein expression was induced at an optical density of 0.6–0.8 with 0.5 mM IPTG and performed for 4–5 h, 30 °C. Cells were harvested at 6.000× *g* for 30 min at room temperature, the pellet was resuspended in 10 mL lysis buffer (NaH_2_PO_4_ (50 mM), NaCl (300 mM), imidazole (10 mM), pH 8) per liter main culture. After a freeze–thaw cycle lysis was performed via sonication, alternating 30 s bursts, 50% intensity with 30 s intervals on ice. The lysate was cleared with centrifugation at 15.000× *g* for 30 min at 4 °C, filtered through 0.45- and 0.2 µm syringe filters and purification was performed using the ÄKTA™ system. After Ni^2+^-affinity chromatography with elution of the protein using an increasing gradient of elution buffer (NaH_2_PO_4_ (50 mM), NaCl (300 mM), imidazole (500 mM), pH 8), His_6_-DTA was reduced by DTT (10 mM) for 40 min at room temperature. Buffer was exchanged to PBS and concentrating was done with Vivaspin^®^ 20, MWCO 5 kDa. Total protein amount was quantified with SDS-PAGE at 6.8 mg/mL with a total yield of 6 mg His_6_-DTA per L main culture. The protein was stored at −80 °C.

#### 4.2.3. Characterization of His_6_-NDPK-A Kinase Activity

**Autophosphorylation assay.** During its nucleoside diphosphate kinase activity, His_6_-NDPK-A itself is autophosphorylated forming a 1-phosphohistidine which allows detection via a specific antibody [39]. His-NDPK-A was incubated with ATP (1 mM) and MgCl_2_ (2 mM) in assay buffer (Tris-HCl (50 mM, pH 8), NaCl (150 mM)) for 5 min. at room temperature. The reaction was terminated with EDTA (5 mM, pH 8), 5 min. at room temperature and addition of cold 5x Laemmli buffer (Tris-HCl (250 mM, pH 8.8), SDS (10%), bromphenol blue (0.02%), glycerol (50%), EDTA (50 mM), DTT (500 mM), PMSF (5 mM)) in a 1:5 ratio. The reaction mix was incubated for 30 min at room temperature and the samples were subjected to SDS-PAGE (gels at pH 8.8) with subsequent western blot analysis. In short, proteins were transferred onto a nitrocellulose membrane. All washing steps were performed with TBS-T (TBS with 0.1% tween-20) and the immunoblotting protocol was based on the manufacturer’s instructions for maximum sensitivity (#34094 and #46641, Thermo Scientific, Waltham, MA, USA). In short, the membrane was incubated with antigen pretreatment solution for 10 min at room temperature, blocked with Roti^®^-Block blocking solution (1× dilution in water, #A151.1, Carl Roth, Karlsruhe, Germany) for 1 h at room temperature and incubated over night at 4 °C with the anti-N1-phosphohistidine antibody (#MABS1341, clone SC50-3, EMD Millipore Corporation, Darmstadt, Germany). The primary antibody was diluted in primary antibody diluent, 120 ng/mL. The secondary antibody HRP-conjugate (#1721019, Bio-Rad, Feldkirchen, Germany) was diluted in blocking solution at 20 ng/mL and incubated for 30 min at room temperature. Detection was performed using the SuperSignal West Femto Maximum Sensitivity Substrate.

**Kinase activity assay.** Generated ATP In Vitro can be quantified via the Kinase-Glo^®^ Luminescent Kinase Assay (Promega, Madison, WI, USA). Purified His_6_-NDPK-A (22.5–360 pM) was incubated with substrate mix containing ADP and GTP at 50 µM each for 30 min. at room temperature in assay buffer (Tris-HCl (50 mM), MgCl_2_ (2 mM), BSA (0.01%), pH 7.5)). Sample size was 12 µL per well, 96-well format (#975075, Greiner Bio-One, Frickenhausen, Germany). As controls samples without substrate, without NDPK-A or buffer alone were included. The kinase reaction was terminated via adding of the Kinase Glo^®^ Luminescent Kinase Assay mix at a 1:1 ratio. After 5–12 min the luminescence intensity was measured at 560 nm.

#### 4.2.4. SDS-PAGE and Western Blotting

Samples were prepared with reducing Laemmli buffer, denatured by heating at 95 °C for 10 min and subjected to SDS-PAGE. By semi-dry blotting, proteins were transferred onto a nitrocellulose membrane (#10600002, GE Healthcare Life Sciences, Uppsala, Sweden) that was then blocked with 5% nonfat dry milk in PBS-T (PBS containing 0.1% Tween-20) for 1 h at room temperature. Primary antibodies were diluted in PBS-T as well and incubated accordingly. Antibodies used were directed against His_6_ (1:1.000, #MA1-21315, Invitrogen, Carlsbad, CA, USA), Hsp90 (1:1.000, #sc-13119, SantaCruz Biotechnology, Dallas, TX, USA) and GAPDH (1:1.000, #sc-365062, SantaCruz Biotechnology, Dallas, TX, USA), all incubated 1 h at room temperature. Anti-NDPK-A (1:200, #sc514515, SantaCruz Biotechnology, Dallas, TX, USA) was applied over night at 4 °C. For detection an anti-mouse IgG kappa binding protein (m-IgGκ BP) HRP-conjugate (1:2.500, #sc-516102, SantaCruz Biotechnology, Dallas, TX, USA) was used, diluted in PBS-T and incubated for 1 h at room temperature. For visualization, the ECL system (#WBKLS0500, EMD Millipore Corporation, Darmstadt, Germany) was used.

#### 4.2.5. Cell Binding Assay

Cells were seeded into a 12-well plate with 150.000 cells/well for MDA-MB-231 cells and 100.000 cells/well for HeLa cells. The cells were adjusted to 4 °C on ice for 30 min., washed twice with ice-cold PBS and incubated with the protein mixtures, 500 µL/well in FCS-free medium for 30 min. on ice. After washing thrice with cold PBS, the supernatant was discarded and following a freeze–thaw cycle, the cells were lyzed in hot 2× Laemmli buffer with DTT (100 mM). Mechanical lysis via pipetting was performed and after boiling the samples were subjected to SDS-PAGE and Western blotting. Cell-bound His_6_-NDPK-A was detected via an anti-His antibody, an anti-mouse HRP-conjugate and the ECL system. To confirm comparable protein loading, Hsp90 was detected. Densitometric analysis was performed with the ImageJ software (National Institutes of Health, Bethesda, MD, USA).

#### 4.2.6. Analysis of Protein Uptake into Cells

Cells were seeded into a 24-well plate with MDA-MB-231 cells 25.000/well and HeLa cells 25.000/well. The protein mixtures were prepared in FCS-containing medium, 250 µL/well and added to the cells for 24 or 48 h, respectively, at 37 °C. In case of the uptake of His_6_-DTA, cells were pre-incubated with BafA1 (30 nM, #sc-201550, SantaCruz Biotechnology, Dallas, TX, USA) for 30 min at 37 °C. BafA1 remained present on the cells throughout the time course of the assay. Cells were washed thrice with cold PBS, the supernatant was discarded and, after a freeze–thaw cycle, lyzed in hot 2× Laemmli buffer with DTT (100 mM). The samples were subjected to SDS-PAGE and subsequent Western blotting. His_6_-proteins were detected with either an anti-His antibody as described above or a specific anti-NDPK-A antibody. Hsp90 was detected for comparison of equal protein loading and band intensity was quantified via the ImageJ software. For the analysis of morphological changes, the cells were monitored throughout the experiment and bright field images were acquired. The percentage of rounded cells was analyzed via blinded/masked quantification using the ImageJ software.

#### 4.2.7. Cell Viability Assay

Cells were seeded into a 96-well plate with MDA-MB-231 cells 2.000/well and HeLa cells 1.600/well. Protein mixtures were added (100 µL/well) as indicated and the cells were incubated at 37 °C, accordingly. As control, either DMSO (20%) or staurosporine (1 µM, S6942, Sigma-Aldrich Chemie, Taufkirchen, Germany) was added to the cells. To determine cell viability, the Cell Titer 96 AQueous One Solution cell proliferation assay [3-(4,5-dimethylthiazol-2-yl)-5-(3-carboxymethoxyphenyl)-2-(4-sulfophenyl)-2H-tetrazolium (MTS) assay] from Promega (#G3580, Promega, Madison, WI, USA) was used. Ten microliters of MTS substrate was added and the cells further incubated for 90 min at 37 °C. The absorbance was measured at 495 nm. Signal intensity was normalized to untreated cells for statistical analysis. In case of the migration assay, a total volume of 25 µL of MTS substrate was added. After 90 min of incubation at 37 °C, 100 µL of each well were transferred into a 96-well plate and the absorbance measured as described above and earlier [32].

#### 4.2.8. Sequential ADP-Ribosylation of EF-2 by DTA

Cells were seeded into 24-well plates (MDA-MB-231 75.000/well, HeLa 50.000/well). The cells were pre-incubated with BafA1 (30 nM) for 30 min at 37 °C followed by addition of His_6_-DTA (5 µg/mL) with or without PA_63_ (1 µg/mL), respectively. After another 12 h incubation at 37 °C the cells were washed 3× with ice-cold PBS, the supernatant was discarded and the cells put to −20 °C. Sample preparation was done as described before for cell-free systems. In brief, lysis was performed on ice in assay buffer (Tris-HCl (20 mM, pH 7.5), EDTA (1 mM), DTT (1 mM), MgCl_2_ (5 mM), Complete (1×)) via scraping of the cells. Lysates were transferred into tubes and incubated with 100 ng His_6_-DTA and 250 pmol biotinylated NAD^+^ (#TRE-4670-500-01, Biozol Diagnostica, Eching, Germany) for 30 min at 37 °C. Hot 5× Laemmli buffer + DTT (100 mM) was added to terminate the reaction (final concentration 1×) and the samples were boiled for 10 min at 95 °C. After SDS-PAGE and Western blotting the biotinylated elongation factor 2 (EF-2) was detected via a streptavidin-peroxidase (#11089153001, Roche Diagnostics, Mannheim, Germany) using the ECL system as described earlier [32]. In addition, GAPDH was detected for loading control.

#### 4.2.9. Immunofluorescence Microscopy

Cells were seeded in 8-well µ-slides (#80826, Ibidi, Martinsried, Germany) with 1.2500 cells/well for MDA-MB-231 and 2.500 cells/well for HeLa. Treatment was performed accordingly, and the cells washed 3 x with warm (37 °C) PBS-T. Cells were fixed with paraformaldehyde (4%) for 20 min at 37 °C and permeabilized with triton ×-100 (0.4%) for 5 min at room temperature, light-excluded. Samples were quenched with warm glycine (100 mM in PBS) for 2 min at room temperature and blocked with 5% nonfat dry milk in PBS-T for 30 min at 37 °C. Immunostaining was done with an anti-His (1:500) primary antibody (see Western blotting) and a goat-anti-mouse-Alexa488 conjugate (1:750, #A-11001, Invitrogen, Carlsbad, CA, USA) for 30 min at 37 °C. In case of additional staining of the filamentous actin with Phalloidin-FITC (1:100, #P5282, Sigma-Aldrich Chemie, Taufkirchen, Germany) for 1 h at room temperature, an Alexa568 conjugate (1:500, #A11004, Invitrogen, Carlsbad, CA, USA) was used. Nuclei were stained with Hoechst 33342 (1:10,000, Thermo Fisher Scientific, Waltham, MA, USA) for 5 min. at room temperature. All dilutions were prepared in 5 % nonfat dry milk in PBS-T. Between each step, five washing steps with warm PBS-T were included. Images were obtained with the iMic digital microscope (FEI, Munich, Germany) and the Live Acquisition 2.6 software (FEI, Munich, Germany) and processed with ImageJ.

#### 4.2.10. Migration Assay (Wound-Healing Assay, Scratch Assay)

The 2-well culture-Insert system for self-insertion from Ibidi was used (#80209, Ibidi, Gräfelfing, Germany). Inserts were placed into wells of a 24-well plate and MDA-MB-231 cells were seeded (25,000 cells per insert-well). The surrounding well area was filled with culture medium. To induce the wound, the insert was removed with forceps and the cells were washed twice carefully with warm culture medium. The protein mixtures were added (250 µL/well) and the cells incubated at 37 °C. Three technical replicates were performed. Migration was monitored via image acquisition; bright field pictures were taken every 4 h. Each well was analyzed with three pictures within the middle of the well and wound/scratch area, respectively. The entire scratch area could be traced. Migration was monitored for up to 48 h with subsequent determination of the cell viability using the MTS assay. For quantitative analysis of the total wound area, the MRI tool was applied using the ImageJ software. Relative wound area was determined in relation to the initial scratch.

#### 4.2.11. Experimental Reproducibility and Statistics

All experiments were independently performed at least three times. The figures show representative results. Immunoblots show sections of larger membrane parts for presentation but the protein bands were always detected on the same membrane, originally. Statistical analysis was performed using the GraphPad Prism software (8.3.1 (549), San Diego, CA, USA).

## Figures and Tables

**Figure 1 ijms-21-03295-f001:**
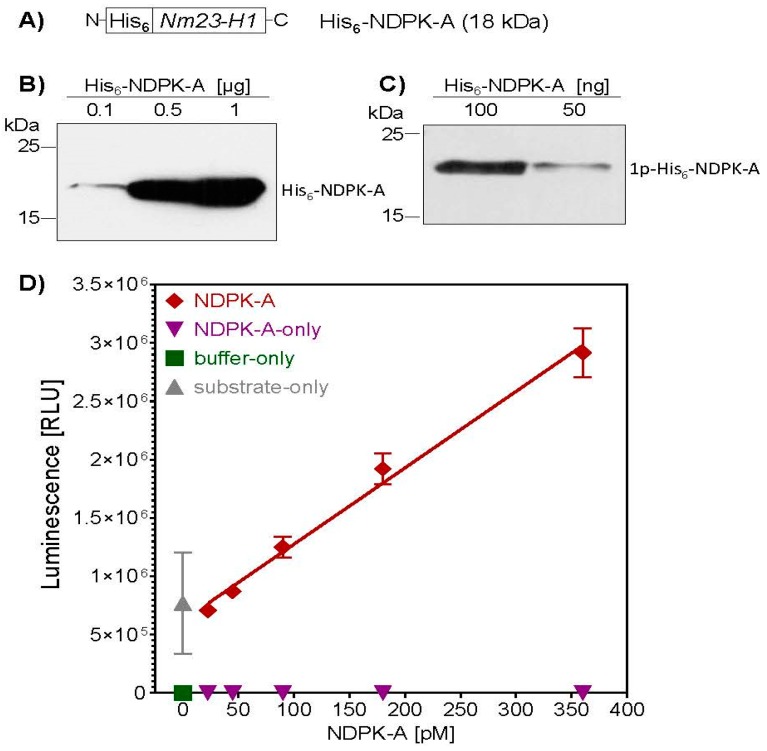
Characterization of recombinant His_6_-NDPK-A. (**A**) After recombinant expression of His_6_-NDPK-A in *E. coli* and purification via affinity chromatography, protein identity was confirmed by Western blotting with a specific anti-NDPK-A antibody (**B**). The kinase activity of His_6_-NDPK-A was analyzed by measuring the proteins autophosphorylation activity (**C**). His_6_-NDPK-A was incubated with ATP (1 mM) and MgCl_2_ (2 mM) for 5 min at room temperature. The reaction was stopped by adding EDTA (5 mM) and the phosphorylated His_6_-NDPK-A intermediate detected by Western blotting with an anti-N1-phosphohistidine (1-pHis) antibody. (**D**) The kinase activity of His_6_-NDPK-A was determined by measuring the substrate conversion status after incubation of increasing amounts His_6_-NDPK-A with substrate for 30 min at room temperature. Controls include His_6_-NDPK-A only, substrate only and buffer only. ATP conversion was quantified with the Kinase-Glo^®^ Luminescent Kinase Assay.

**Figure 2 ijms-21-03295-f002:**
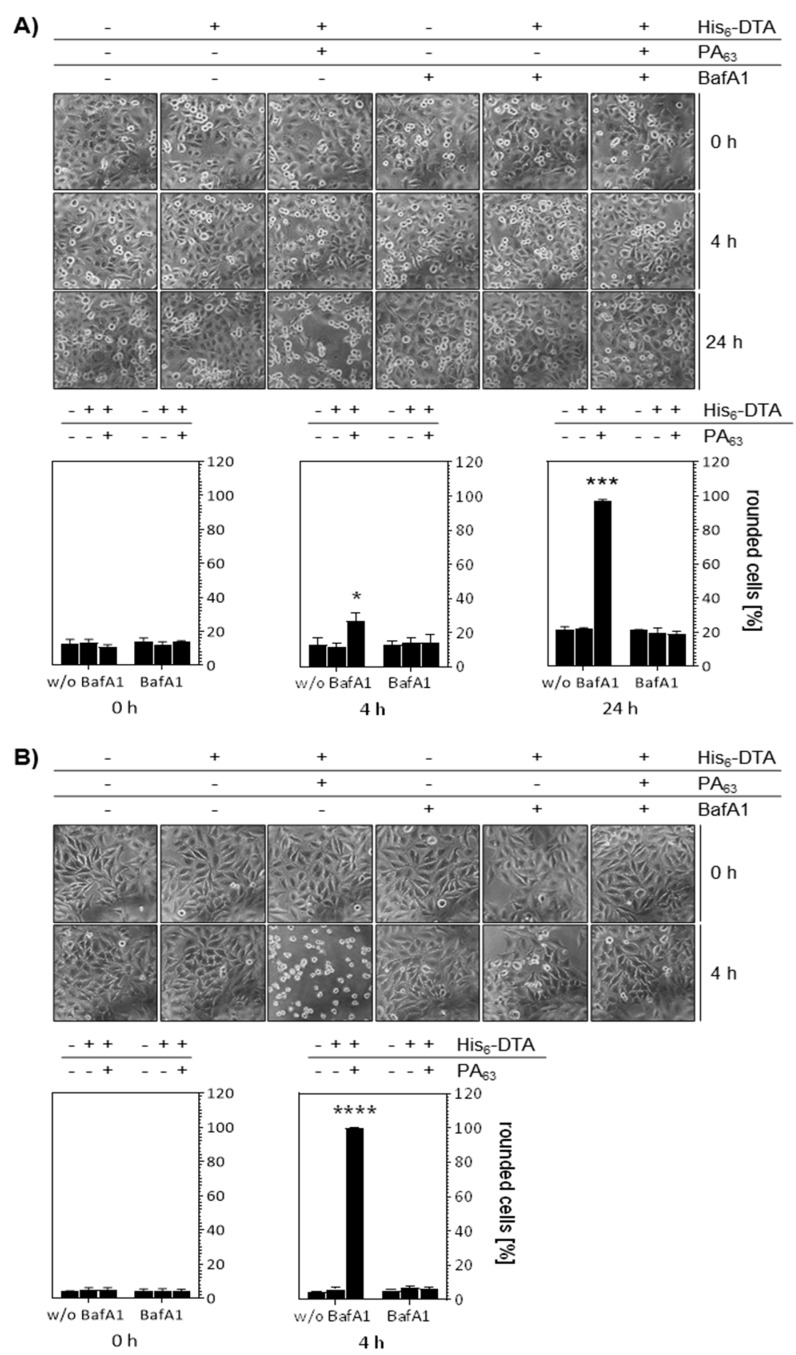
MDA-MB-231 and HeLa cells respond to treatment with His_6_-DTA and PA_63_ with cell rounding. MDA-MB-231 (**A**) as well as HeLa (**B**) cells were incubated with BafA1 (30 nM) for 30 min at 37 °C. Then, His_6_-DTA (5 µg/mL) with or without PA_63_ (1 µg/mL) was added and cells were further incubated at 37 °C for 24 h. Representative pictures are shown before application of the proteins and after 4 and 24 h of treatment. The number of rounded cells at the indicated time points were quantified using ImageJ and the data is given as mean ± SD, n = 3. Significance was tested using One-Way ANOVA followed by Dunnett’s multiple comparison test. * *p* ≤ 0.05; *** *p* ≤ 0.001, **** *p* ≤ 0.0001 vs. untreated controls.

**Figure 3 ijms-21-03295-f003:**
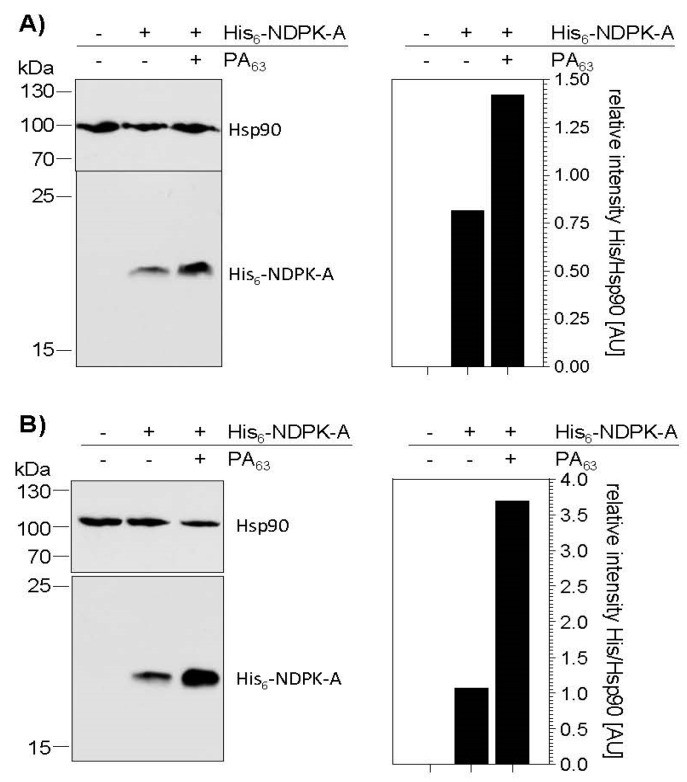
PA_63_ mediates binding of His_6_-NDPK-A to MDA-MB-231 and HeLa cells. After adjusting MDA-MB-231 (**A**) and HeLa (**B**) cells to 4 °C they were incubated with His_6_-NDPK-A (30 µg/mL) with or without PA_63_ (3 µg/mL) for 30 min on ice. After washing, the amount of His_6_-NDPK-A bound to the cells was determined by Western blotting with an anti-His-antibody. Hsp90 was detected to confirm comparable protein loading. Densitometric analysis was performed using the ImageJ software. Signal intensity of His_6_-NDPK-A was quantified against the Hsp90 signal.

**Figure 4 ijms-21-03295-f004:**
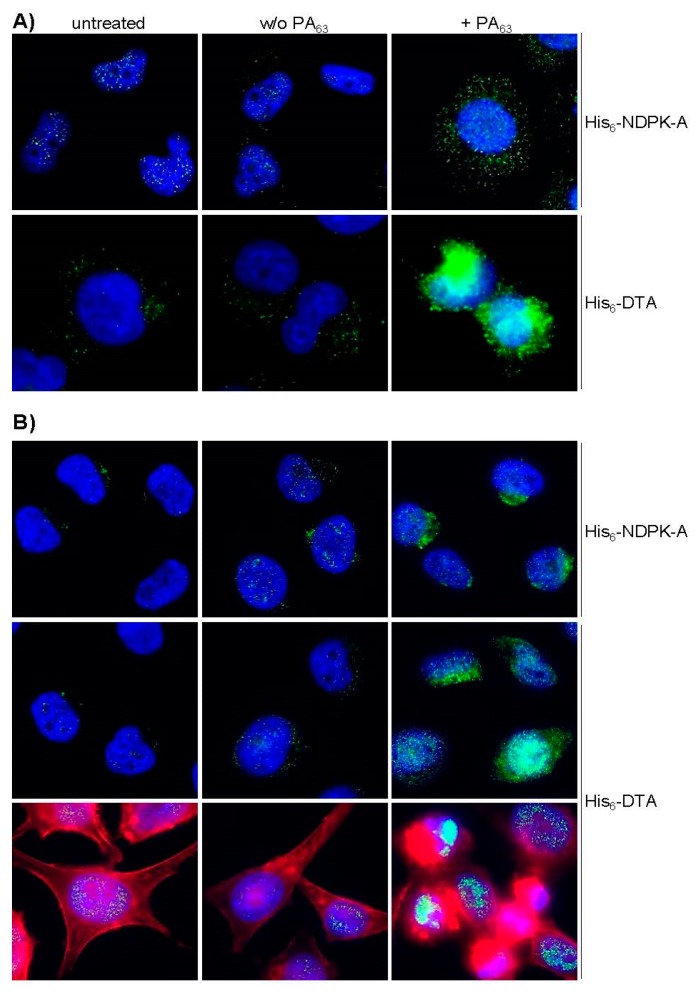
PA_63_ mediates the delivery of His_6_-NDPK-A and His_6_-DTA into MDA-MB-231 and HeLa cells. (**A**) MDA-MB-231 cells were incubated for 48 h at 37 °C with either His_6_-NDPK-A (10 µg/mL) or His_6_-DTA (5 µg/mL) in the presence or absence of PA_63_ (1 µg/mL). (**B**) HeLa cells were treated the same way for 3 h. His-tagged proteins (green) were visualized by fluorescence microscopy. After incubation with an anti-His-antibody, immunostaining was performed with a goat anti-mouse-Alexa Fluor568 antibody. Nuclei were stained with Hoechst 333342 (blue); F-actin cytoskeleton was stained with Phalloidin-FITC (red) to visualize the His_6_-DTA-induced cell-rounding. Representative cells are shown.

**Figure 5 ijms-21-03295-f005:**
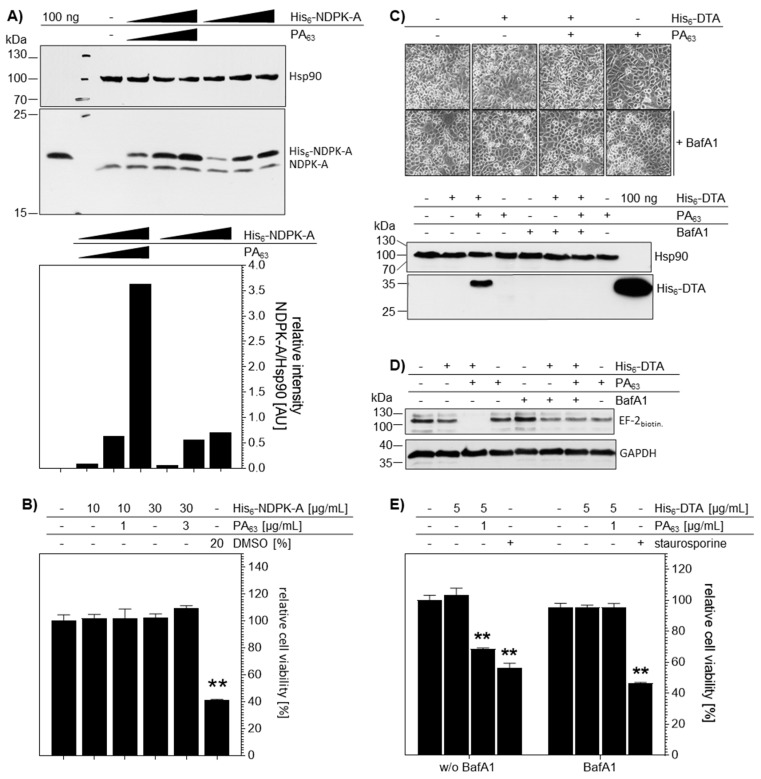
PA_63_ mediates delivery of His_6_-NDPK-A and His_6_-DTA into MDA-MB-231 cells. (**A**) PA_63_ delivers His_6_-NDPK-A into MDA-MB-231 cells in a concentration-dependent manner. After incubation of the cells with His_6_-NDPK-A and PA_63_ (10 + 1, 18 + 6 or 30 + 3 µg/mL) or His_6_-NDPK-A only (10, 18 or 30 µg/mL) for 48 h at 37 °C Western blot analysis was performed. The uptake of His_6_-NDPK-A (18 kDa) was detected with the specific anti-nm23-H1 antibody. The lane on the far left contains purified recombinant His_6_-taged NDPK-A only which was detected at 18 kDa. Endogenous NDPK-A is detected in the cell lysates at 17 kDa. Hsp90 was detected as loading control and densitometric analysis was performed using the ImageJ software. The signal intensity of His_6_-NDPK-A was quantified against Hsp90. (**B**) Cell viability is not affected by the treatment of cells with His_6_-NDPK-A and PA_63_. After incubation of the cells with His_6_-NDPK-A and PA_63_ or His_6_-NDPK-A-only for 48 h at 37 °C, cell viability was determined by the MTS assay. Data were normalized to control (untreated cells). The known reduction by 20% (by vol.) DMSO was used as control. Data are given as mean ± SD, n = 3. (**C**) Uptake of His_6_-DTA is mediated by PA_63_ and can be blocked by BafA1. Cells were incubated with BafA1 (30 nM) for 30 min at 37 °C followed by 12 h incubation with His_6_-DTA (5 µg/mL) with or without PA_63_ (1 µg/mL). Representative pictures depict no morphological changes after incubation and before sample preparation. His_6_-DTA was detected by an anti- His_6_-antibody. Detection of Hsp90 served as loading control. (**D**) PA_63_ mediates cytosolic localization His_6_-DTA in MDA-MB-231 cells. After incubation with BafA1 (30 nM) for 30 min, His_6_-DTA (5 µg/mL) with or without PA_63_ (1 µg/mL) was added for 12 h at 37 °C. After cell lysis, In Vitro ADP-ribosylation was performed using biotinylated NAD^+^ (10 mM) as substrate. Modified proteins were precipitated by strep-POD and biotinylated EF-2 was detected by immunoblotting. GAPDH expression was used as loading control. (**E**) Cell viability is decreased after treatment of cells with His_6_-DTA and PA_63_. Cell viability was determined via the MTS assay after 24 h of incubation at 37 °C with His_6_-DTA with or without PA_63_. Staurosporine (1 µM) was used as positive control. Data are normalized to untreated controls and given as mean ± SD, n = 3. Significance was tested using One-Way ANOVA followed by Dunnett’s multiple comparison test. ** *p* ≤ 0.01 vs. untreated controls.

**Figure 6 ijms-21-03295-f006:**
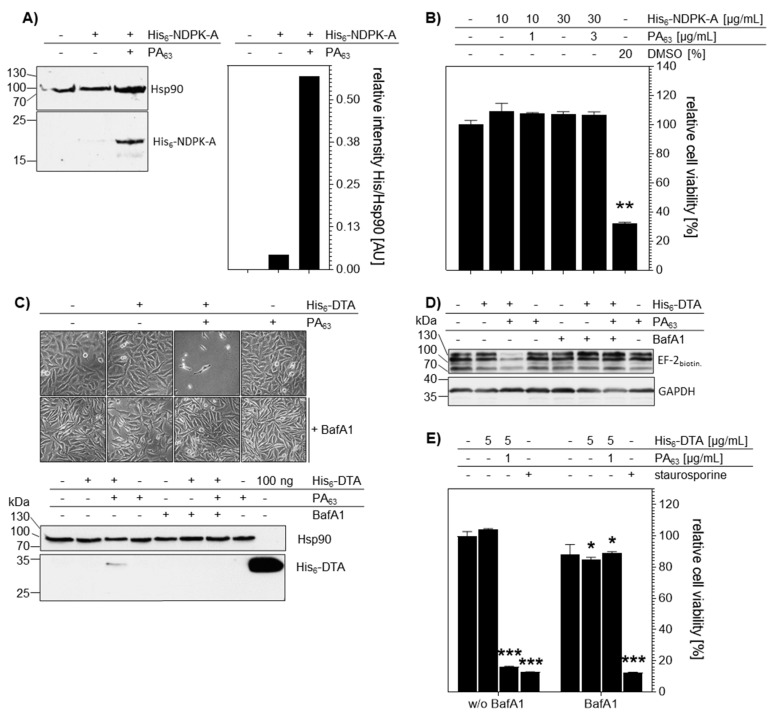
PA_63_ mediates delivery of His_6_-NDPK-A and His_6_-DTA into HeLa cells. (**A**) PA_63_ delivers His_6_-NDPK-A into HeLa cells in a PA_63_-dependent manner. After incubation of the cells with His_6_-NDPK-A (10 µg/mL) and PA_63_ (1 µg/mL) or His_6_-NDPK-A only for 6 h at 37 °C, the cells were analyzed by immunoblotting. The uptake of His_6_-NDPK-A was detected with an anti-His-antibody. Hsp90 was detected as loading control and densitometric analysis performed using the ImageJ software. Signal intensity of His_6_-NDPK-A was quantified against Hsp90. (**B**) Cell viability is not reduced by treatment of cells with His_6_-NDPK-A plus PA_63_. After incubation of the cells with His_6_-NDPK-A plus PA_63_ or His_6_-NDPK-A alone for 48 h at 37 °C, the cell viability was determined by the MTS assay. Data were normalized to untreated cells (control). DMSO served as positive control. Data are given as mean ± SD, n = 3. (**C**) Uptake of His_6_-DTA is mediated by PA_63_ and is sensitive to BafA1. Cells were incubated with BafA1 (30 nM) for 30 min at 37 °C followed by 24 h incubation with His_6_-DTA (5 µg/mL) with or without PA_63_ (1 µg/mL). Representative pictures are shown. Thereafter, cell lysates were analyzed by immunoblotting for the presence of His_6_-DTA with an anti-His-antibody. Detection of Hsp90 was used as loading control. (**D**) PA_63_ mediates delivery of His_6_-DTA into the cytosol of HeLa cells. Substrate modification was determined via strep-POD after sequential ADP-ribosylation of EF-2 as described. To confirm comparable protein loading, GAPDH was again used as loading control. (**E**) Cell viability is decreased after treatment of cells with His_6_-DTA plus PA_63_. Cells were incubated with His_6_-DTA alone or in combination with PA_63_ for 24 h at 37 °C. Cell viability was determined by the MTS assay. Staurosporine (1 µM) served as positive control. Data are given as mean ± SD, n = 3. Significance was tested using One-Way ANOVA followed by Dunnett’s multiple comparison test. * *p* ≤ 0.05, ** *p* ≤ 0.01, *** *p* ≤ 0.001 vs. untreated controls.

**Figure 7 ijms-21-03295-f007:**
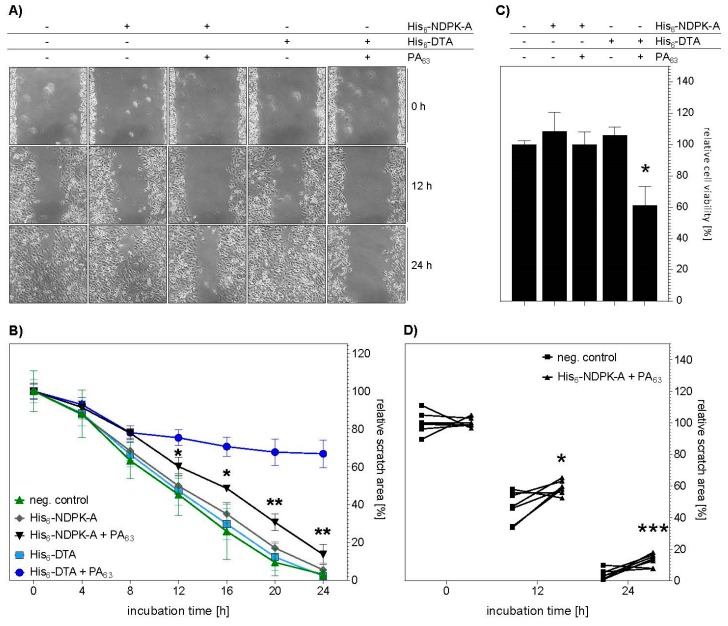
Uptake of His_6_-NDPK-A decreases the migration of MDA-MB-231 cells. (**A**) 24 h before treatment MDA-MB-231 cells were seeded into the wells of two-well ibidi culture inserts. After removal of the inserts to create a wound (scratch), His_6_-NDPK-A (30 µg/mL) plus PA_63_ (3 µg/mL), His6-NDPK-A alone (30 µg/mL), His_6_-DTA (5 µg/mL) plus PA_63_ (1 µg/mL) or His_6_-DTA alone (5 µg/mL) were added to the cells. The cells were incubated at 37 °C and after the indicated time points, pictures were acquired. Assay triplicates were performed. Three areas of the wound of each well were monitored and representative pictures are shown. (**B**) Quantitative analysis was performed via the ImageJ software to determine the relative surface of the wound area before and after treatment with His-NDPK-A and PA_63_. For each time point significance was tested by One-Way-ANOVA followed by Dunnett’s multiple comparison test. * *p* ≤ 0.05, ** *p* ≤ 0.01 vs. untreated controls, n = 3. (**C**) Cell viability was determined by MTS assay after 48 h of incubation at 37 °C. (**D**) Data from three independent experiments with a total of seven replicates were analyzed after 12 and 24 h of treatment. Data are presented as scatter plots to visualize the individual replicates Statistical significance was tested via a modified t-test using the two-stage step-up method of Benjamini, Krieger and Yekutieli, which does not assume equal distribution. * *p* ≤ 0.05, *** *p* ≤ 0.001 vs. untreated cells, n = 7.

**Figure 8 ijms-21-03295-f008:**
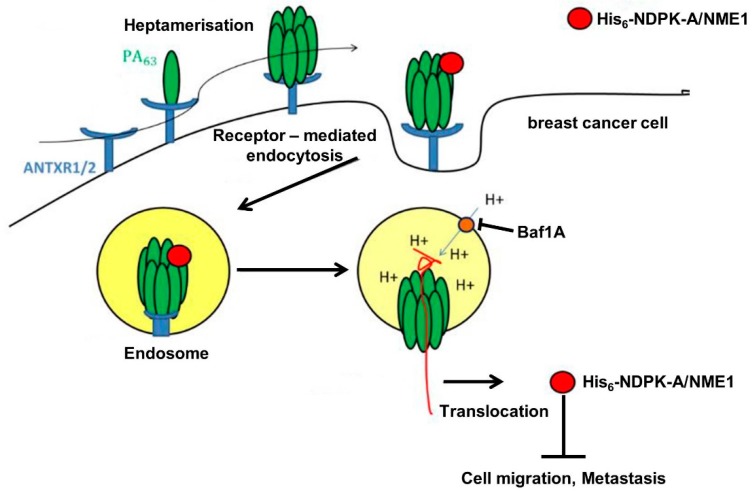
Schematic representation of the PA63-mediated transport of recombinant His-tagged NDPK-A/NME1 into the cytosol of human cancer cells. Explanations are given in the text. ANTXR1/2, anthrax toxin receptors 1 and 2; Baf A1, bafilomycin A1.

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
