# Peer review of "Bacillus anthracis’ PA63 Delivers the Tumor Metastasis Suppressor Protein NDPK-A/NME1 into Breast Cancer Cells"

_ijms, 2020, doi:10.3390/ijms21093295_

Round 1
Reviewer 1 Report
In this paper Felix and coworkers aimed to establish the Bacillus anthracis 18 protein toxins´ transport component PA63 as transporter for the delivery of His-tagged human 19 NDPK-A into the cytosol of cultured cells including human MDA-MB-231 breast cancer cells. The investigation is well conducted, thoroughly described and it produced promising results. The manuscript is ready for publication after addressing a few minor points:
- It would be beneficial for readers to have a schematic representation of the delivery system described, to be able to follow the paper more easily.
- The F-actin image is rather blurry. Although, in this case, the F-actin staining is merely here to visualize the transformation of the cell shape it would be good to replace it with a better one, if possible.
- The authors proved the transport of NME1 in the cytoplasm. Did you try to track the possible transport into other compartments after the mentioned procedure? It is known that NME1 normally sometimes occupy the nucleus or attach to the endoplasmic reticulum. Does this imply that its antimigratory activity comes exclusively from its actions in the cytosol? Please comment.
Author Response
In this paper Felix and coworkers aimed to establish the Bacillus anthracis 18 protein toxins´ transport component PA63 as transporter for the delivery of His-tagged human 19 NDPK-A into the cytosol of cultured cells including human MDA-MB-231 breast cancer cells. The investigation is well conducted, thoroughly described and it produced promising results. The manuscript is ready for publication after addressing a few minor points:
We thank this reviewer very much for the time spent upon our behalf and fort he very positive comments about our study. We addressed the minor points raised by this reviewer as follows:
It would be beneficial for readers to have a schematic representation of the delivery system described, to be able to follow the paper more easily.
Thank you very much for this excellent suggestion. The new Figure 8 on page 13 with an explanatory legend schematically demonstrates the uptake of the His-tagged NDPKA/NME1 protein by the PA63 delivery system.
The F-actin image is rather blurry. Although, in this case, the F-actin staining is merely here to visualize the transformation of the cell shape it would be good to replace it with a better one, if possible.
We absolutely agree, but unfortunately, we did not obtain better pictures here. As this reviewer points out, the actin staining was only performed to confirm the alteration of the cell shape. This is quite obvious also in used pictures.
The authors proved the transport of NME1 in the cytoplasm. Did you try to track the possible transport into other compartments after the mentioned procedure? It is known that NME1 normally sometimes occupy the nucleus or attach to the endoplasmic reticulum. Does this imply that its antimigratory activity comes exclusively from its actions in the cytosol? Please comment.
This is an excellent comment and a very good suggestion, but we did not perform such analyses so far. Therefore, we can not exclude that a certain portion of the internalized NME1 protein translocates from the cytosol into the nucleus. However, we feel that this highly interesting topic is beyond the focus of the present study which was conducted to proof principally that delivery of NME1 into the cytosol of highly metastatic breast cancer cell by a toxin-based transport system reduces the migration of these cells. Of course, we will further look into this in our ongoing continuation study.
We tried to address all points raised carefully and thus hope that our revised manuscript is now acceptable for publication in the IJMS.
Reviewer 2 Report
Bacillus anthracis´ PA63 delivers the tumor metastasis 2 suppressor protein NDPK-A/NME1 into breast cancer 3 cells, IJMS_795765
The manuscript from Felix et al. dissects the role NME1 in the migratory ability of human breast cancer cells via the delivery of NME1 by PA63
The authors focus on a scientifically important field. Structure of the paper is logic and well organized. Due to the high mortality rate of solid malignancies in adults the manuscript fits in the scope of IJMSjournal.
I have the following major comments:
- Can be PA63 immunogenic and cause severe immune reaction harmful to the patient or otherwise can it be advantageous to boost antitumor immunity as an adjuvant?
- Has the nme1 transcript lower half life in highly metastatic cells possibly regulated by a miRNA or the NME1 protein is degraded e.g. via ubiquitination in highly metastatic cells? This is relevant question to consider the half life of the exogenous NME1 as a clinical application.
- Correct the legends of Figure 1 C and D
- Quantitation of the confluency of replicates from the images could help to perform statistics on the effect in Figure 2. Alternatively, the xCelligence system would be a nice tool to monitor this cell rounding.
- Why not FACS, native cell surface staining with flow cytometry was used to monitor cell bound (at 4 °C) His6-NDPK-A? An immunofluorescent microscopy picture should be presented (incubated at 4 °C) as a supplement to rule out the intracellular signal in Figure 3.
- The “only secondary antibody control” should be presented as an insert or supplement corresponding to Figure 4.
Author Response
Bacillus anthracis´ PA63 delivers the tumor metastasis 2 suppressor protein NDPK-A/NME1 into breast cancer 3 cells, IJMS_795765
The manuscript from Felix et al. dissects the role NME1 in the migratory ability of human breast cancer cells via the delivery of NME1 by PA63
The authors focus on a scientifically important field. Structure of the paper is logic and well organized. Due to the high mortality rate of solid malignancies in adults the manuscript fits in the scope of IJMSjournal.
I have the following major comments:
We also thank this reviewer very much for the time and the positive evaluation of our manuscript. In detail, we addressed the points as follows:
Can be PA63 immunogenic and cause severe immune reaction harmful to the patient or otherwise can it be advantageous to boost antitumor immunity as an adjuvant?
This is a very important question and this topic was addressed earlier mainly by the research groups around R. John Collier and Stephan Leppla, who developed PA as transport system for therapeutic proteins into human cells. In brief, PA was originally used as a vaccine for soldiers (protective antigen) but the immunogenic effect was not so strong. When used for therapeutic applications, there were no severe adverse effects due to immunogenic effects caused by PA-based transport systems in vivo (e.g. Jack, S.; Madhivanan, K.; Ramadesikan, S.; Subramanian, S.; Edwards, D.F.; Elzey, B.D.; Dhawan, D.; McCluskey, A.; Kischuk, E.M.; Loftis, A.R.; et al. A novel, safe, fast and efficient treatment for Her2-positive and negative bladder cancer utilizing an EGF-anthrax toxin chimera. Int. J. Cancer 2020, 146, 449–460.). We cite this paper in our manuscript and mention this now in the discussion section on page 14, ln. 400/401.
Has the nme1 transcript lower half life in highly metastatic cells possibly regulated by a miRNA or the NME1 protein is degraded e.g. via ubiquitination in highly metastatic cells? This is relevant question to consider the half life of the exogenous NME1 as a clinical application.
This is an highly interesting point but we did not address this aspect in our present proof-of-concept study where we focused on the delivery of NME1 by the toxin-based transport system. This will also be addressed in the ongoing continuation study.
Correct the legends of Figure 1 C and D
Thank you, done. Figure references were adjusted in the results section on page 3 lines 104, 107, 109 and brackets-placement in the figure legend in the lines 113, 115 and 119.
Quantitation of the confluency of replicates from the images could help to perform statistics on the effect in Figure 2. Alternatively, the xCelligence system would be a nice tool to monitor this cell rounding.
Thank you for this excellent suggestion. Although we could not use this specific approach, we included now a quantitative analysis into Figure 2 on page 5 to emphasize the visual analysis. See changes on page 4 line 143; page 5, Figure 2, legend lines 151 – 154; page 17 material and methods part 4.3.6., lines 527/528.
Why not FACS, native cell surface staining with flow cytometry was used to monitor cell bound (at 4 °C) His6-NDPK-A? An immunofluorescent microscopy picture should be presented (incubated at 4 °C) as a supplement to rule out the intracellular signal in Figure 3.
We tried to perform such experiments but had to stop them due to technical problems. We learned that extensive washing was necessary to remove non-specifically bound His-NDPK-A from the cell surface. Unfortunately, the binding at 4°C the attachment of His-NDPK-A to PA63 is also very weak and therefore, the bound protein was similarly removed by the required washing steps. Therefore, we did not continue to analyze the binding under such conditions. As we and others characterized the PA63-dependent transport of His-tagged cargo proteins in detail for various other proteins and demonstrated the specific binding and uptake herein, we believe that such a picture is dispensible.
The “only secondary antibody control” should be presented as an insert or supplement corresponding to Figure 4.
We did such controls many times earlier for different cell types and never obtained relevant non-specific signals by using the secondary antibody alone. We did however not perform this control in the shown experiment. Due to the shut down of our labs during the ongoing Covid-19 pandemic, we are currently unable to provide such a picture.
We tried to address all points raised carefully and thus hope that our revised manuscript is now acceptable for publication in the IJMS.